# In Vitro Hydrolytic Degradation of Polyester-Based Scaffolds under Static and Dynamic Conditions in a Customized Perfusion Bioreactor

**DOI:** 10.3390/ma15072572

**Published:** 2022-03-31

**Authors:** Pilar Alamán-Díez, Elena García-Gareta, Pedro Francisco Napal, Manuel Arruebo, María Ángeles Pérez

**Affiliations:** 1Multiscale in Mechanical and Biological Engineering, Instituto de Investigación en Ingeniería de Aragón (I3A), Instituto de Investigación Sanitaria Aragón (IIS Aragón), University of Zaragoza, 50018 Zaragoza, Spain; garciage@posta.unizar.es (E.G.-G.); pedro_napal@hotmail.com (P.F.N.); angeles@unizar.es (M.Á.P.); 2Division of Biomaterials and Tissue Engineering, UCL Eastman Dental Institute, University College London, London WC1E 6BT, UK; 3Instituto de Nanociencia y Materiales de Aragón (INMA), Consejo Superior de Investigaciones Científicas (CSIC), University of Zaragoza, 50018 Zaragoza, Spain; arruebom@unizar.es; 4Department of Chemical Engineering, University of Zaragoza, Campus Río Ebro-Edificio I + D, C/Poeta Mariano Esquillor S/N, 50018 Zaragoza, Spain

**Keywords:** biomaterials, scaffolds, degradation, hydrolysis, polyester-based materials, flow perfusion

## Abstract

Creating biofunctional artificial scaffolds could potentially meet the demand of patients suffering from bone defects without having to rely on donors or autologous transplantation. Three-dimensional (3D) printing has emerged as a promising tool to fabricate, by computer design, biodegradable polymeric scaffolds with high precision and accuracy, using patient-specific anatomical data. Achieving controlled degradation profiles of 3D printed polymeric scaffolds is an essential feature to consider to match them with the tissue regeneration rate. Thus, achieving a thorough characterization of the biomaterial degradation kinetics in physiological conditions is needed. Here, 50:50 blends made of poly(ε-caprolactone)–Poly(D,L-lactic-co-glycolic acid (PCL-PLGA) were used to fabricate cylindrical scaffolds by 3D printing (⌀ 7 × 2 mm). Their hydrolytic degradation under static and dynamic conditions was characterized and quantified. For this purpose, we designed and in-house fabricated a customized bioreactor. Several techniques were used to characterize the degradation of the parent polymers: X-ray Photoelectron Spectroscopy (XPS), Gel Permeation Chromatography (GPC), Scanning Electron Microscopy (SEM), evaluation of the mechanical properties, weigh loss measurements as well as the monitoring of the degradation media pH. Our results showed that flow perfusion is critical in the degradation process of PCL-PLGA based scaffolds implying an accelerated hydrolysis compared to the ones studied under static conditions, and up to 4 weeks are needed to observe significant degradation in polyester scaffolds of this size and chemical composition. Our degradation study and characterization methodology are relevant for an accurate design and to tailor the physicochemical properties of polyester-based scaffolds for bone tissue engineering.

## 1. Introduction

According to the World Health Organization, over 500,000 bone grafting procedures are performed each year, becoming the second most commonly transplanted tissue after blood [1]. Autologous bone grafts harvested from healthy bone can be considered as the gold standard. Nevertheless, it may cause donor site morbidity, low mechanical performance, long recovery periods and require a complex graft shaping process. In this context, bone tissue engineering (BTE) may be seen as a promising solution. Bone graft substitutes (BGS) are synthetic biomaterials that are implanted in a bone defect. They aim to provide temporary functions while bone is healing at the same time that the graft is being degraded [2]. Thus, BGS should be suitable for an appropriate bone cell growth, provide an osteoconductive and osteoinductive environment, and their geometric design must mimic the hierarchical and complex bone porous structure [3,4].

Different biomaterials have been investigated in BTE, such as ceramics, titanium and biodegradable polymers. Scaffolds fabricated from ceramics such as tricalcium phosphate or hydroxyapatite are osteoconductive and have mechanical properties comparable to those of native bone [5,6]. However, ceramic scaffolds lack interconnected pores, are complicated to customize and are prone to fracture. Metals such as titanium and its alloys fabricated by additive manufacturing [7] or as titanium foams [8] may also be used as bone scaffolds. Nevertheless, titanium is non-degradable, and thus would constitute a permanent implant, requiring in some cases a secondary or removal surgery. As a consequence, biodegradable polymers have acquired substantial interest owing to their wide range of applications in BTE despite their reduced mechanical resistance. Chandra et al. (2020) thoroughly reviewed the wide variety of bone polymeric scaffolding [9]. Among other biodegradable synthetic polymers, polyesters like poly(glycolic acid) (PGA), poly(lactic acid) (PLA), their copolymer PLGA and poly(ε-caprolactone) (PCL) are commonly used as scaffolds for tissue engineering, as carriers for drug-delivery applications and as orthopedic fixation devices [10]. These polyesters are often cheaper than natural polymers such as collagen, can be fabricated with a tailored architecture, possess long shelf life and are produced in large scale. Their degradation characteristics can be controlled by varying both the blend polymer itself and the individual constitutive polymers [11]. Besides this, these materials can contain organic or inorganic phases of native and artificial bone (e.g., synthetic hydroxyapatite) as fillers that offer osteoconductivity, toughness and strength, which make them ideal materials to recreate bone structure [12].

The latest advances in manufacturing processes may improve polymeric BGS development. Previous works have studied degradation [13], cell interaction [14] and drug delivery [15] using PCL and PLGA electrospun scaffolds. Nevertheless, this manufacturing technique is not suitable for certain bone tissue applications due to their different mechanical properties compared to natural bone, such as the stiffness or structural rigidity. Apart from that, electrospinning does not allow a complete control over porosity and pore size and geometry [16]. To overcome these issues, other additive manufacturing techniques can be introduced in BTE, allowing customized scaffold fabrication. Hence, other rapid prototyping technologies such as 3D printing are becoming increasingly popular [17,18]. The 3D printing technologies fabricate constructs via layer by layer and their final geometry is dictated and thoroughly guided by a computer-aided design (CAD) model. Thus, complex structures using bioactive materials may be created. In addition, by converting medical images of bone defects into CAD models, patient-specific implants can be printed and implanted [19].

Degradation is a crucial material property to be tailored when using polyester-based scaffolds in order to allow new bone tissue formation whilst the scaffold is degrading at the same rate as the new tissue forms. Polyester degradation in vivo implies, among other degradation mechanisms, a chemical hydrolysis reaction of its ester bonds [20]. Thus, the scaffold properties: monomer structure, molecular weight, co-polymer ratio and crystallinity are modified while they are hydrolyzed [21]. The external environment has an effect and is itself affected by the hydrolysis process in terms of pH, temperature and presence/absence of enzymes [22]. It is important to elucidate the structural changes of the polymeric scaffolds during this degradation process, as well as to measure their degradation rate. Ultimately, the degradation rate should be comparable to that of bone forming rate, which is a multiple factor dependent process in which several issues are involved, such as the presence of damaged bone and its mechanical environment, defect size or applied drugs to name a few. Despite the importance of the degradation properties of polyester-based scaffolds, few studies report on them or thoroughly characterize them.

The main aim of this work was the establishment of a methodology for the degradation characterization and quantification of polyester-based scaffolds. Two conditions were analyzed: static and dynamic by flow perfusion, where the later may closely resemble physiological conditions. The 3D printed PCL-PLGA blends (50:50 *w/w*) were selected as test materials due to their biocompatibility, distinct mechanical features and degradation rate. PLGA has a fast degradation rate, which can also be tuned depending on its molecular weight and its lactic acid–glycolic acid ratio, but low mechanical properties. On the other hand, PCL shows a slow degradation rate, but considerable mechanical resistance. Both features prompted PCL-PLGA use for long term applications in bone repair [23]. We hypothesized that degradation of 3D printed polyester scaffolds under perfusion flow is faster than that under static conditions. Scaffold degradation was monitored for two and four weeks by measuring sample weight loss ($W_{L}$), the weight-average molecular weight ($M_{W}$) and polydispersity index (PI), the amount of O-C=O bonds on the surface related to the number of terminal acid groups, changes in C/O ratios, changes in its elastic modulus and mechanical properties, the evolution of the scaffolds pore size, and indirectly by the change in the pH of the phosphate buffered saline (PBS) used as test medium. To achieve our aim, we designed a customized perfusion bioreactor which allowed the study of both static and dynamic conditions. The application of flow perfusion is generally lacking in BGS studies prior to in vivo implantation. The bioreactor allows the assessment of four samples simultaneously and the standard interface and connection system allow different connection configurations to design a customized experiment. Thus, the bioreactor presented here could be a useful tool for the in vitro characterization of BTE scaffolds.

## 2. Materials and Methods

### 2.1. Co-Polymer Fabrication and Sample Preparation

PCL with an average molecular weight ($M_{W}$) of 45 kDa and PLGA Resomer^®^ RG 502 H, $M_{W}$ of 7 to 17 kDa, were purchased from Sigma-Aldrich (St. Louis, MO, USA). Both polymers (Figure 1) were used as received without any further modification. Then, the PCL-PLGA blend (50:50 wt) was fabricated by dissolving PCL pellets and PLGA powder in dichloromethane (DCM), and a subsequent casting and solvent evaporation. Afterwards, blend filaments were extruded with an in-house mechanical extruder. Then, cylindrical scaffolds were printed (diameter 7 mm, height 2 mm, space between fibers 400 μm) by plotting fibers with 45${}^{\circ }$ angle-steps between two successive layers by using an Original Prusa i3 MK2/S 3D printer. In Figure 2B (number 4), a simple sketch of the printed scaffold can be seen. Once fabricated, scaffolds were sterilized by UV irradiation, soaked in PBS and incubated at 37 ${}^{\circ }$C for two or four weeks. Scaffold degradation was quantified in an in-house bioreactor with and without perfusion flow as we mentioned before.

Each experiment was conducted 3 times. Since four samples were assayed per experiment, each condition held 12 repetitions. Incubated samples were divided at the end of each experiment to perform different analyses: 4 samples to GPC, 4 to SEM and 4 to XPS and, subsequently, mechanical testing (*n* = 4). In order to compare with non-degraded scaffolds, just-printed samples were used as controls: Just-printed scaffolds were sterilized and stored at 4 ${}^{\circ }$C during 2 and 4 weeks to be used as controls for the weight loss determination (Section 2.3). PBS stored in the incubator (37 ${}^{\circ }$C, 21% O${}_{2}$ and 5% CO${}_{2}$) in 24-well plates was used as control for pH monitoring (Section 2.4). Samples just printed were tested and used as control for surface oxidation (Section 2.5) and to evaluate the mechanical properties of the pristine materials (Section 2.6).

### 2.2. Degradation Experimental Set Up

To test PCL-PLGA-based scaffolds degradation, a new bioreactor was designed and in-house fabricated (Figure 2A). The bioreactor contained four individual chambers and they formed four closed circuits interfaced with 1.42 mm outside diameter tubes (Tygon^®^ S3 ™E-LFL, VWR, Lutterworth, UK). In Figure 2B, a schematic design of the bioreactor interfaced system can be observed.

Prior to the whole system set up, every component of the bioreactor was sterilized via autoclave (Tuttnauer, Breda, The Netherlands) at 120 ${}^{\circ }$C for 45 min. Then, each circuit was completely filled with PBS (total volume 2 mL each). Perfusion tests were done inside the bioreactor: a roller pump Hei-FLOW Precision (Heidolph Instruments GmbH & Co., Schwabach, Germany) was used to impose a PBS flow rate of 4 mL/min [24]. Samples were incubated in normoxia (21% O${}_{2}$ and 5% CO${}_{2}$) for two and four weeks. Degradation under static conditions was also conducted inside the bioreactor with no pump working.

### 2.3. Weight Loss Evaluation

Before sterilization, scaffolds were weighted to determine their initial weight ($W_{i}$) (RADWAG-MYA5.4Y microbalance, 1 μg readability). Then, samples were sterilized by UV irradiation and the experiment was initialized. Control scaffolds also underwent UV exposure prior to their storage at 4 ${}^{\circ }$C in a dry environment. After the whole experiment period, samples were taken from the PBS and washed twice with 2 mL of deionized water for 5 min each time. Samples were dried before weighting them to get their final weight ($W_{f}$). Equation (Equation 1) was used to evaluate the overall mass change calculated from the initial and final values.
(1)$$WL\left(\%\right)=\frac{W_{i}-W_{f}}{W_{i}}\times 100$$

### 2.4. pH Variation

For every experiment, PBS was exchanged every 48 h and the pH of collected PBS measured (Fisher Scientific Accumet AE150 pH meter). Sterile 3 mL-syringes (BD Medical, Franklin Lakes, NJ, USA) were used to draw in and out the medium. Each circuit contained 2 mL of PBS and no important evaporation was observed after 48 h.

### 2.5. Gel Permeation Chromatography

Absolute molecular weights (M${}_{W}$) and the polydispersity index (PI) of the polymers were determined by Gel Permeation Chromatography (GPC). These analyses were carried out using a Waters 2695 instrument equipped with three PLGel Mixed C (7.80 × 300 mm) columns and a Wyatt three-detector setup (Minidaw TREOS^®^ (MALS), Optilab Rex^®^ 10 (DRI) and ViscostarII^®^ Viscometer; Wyatt Technology: Goleta, CA, USA).

The samples were eluted with tetrahydrofuran (THF) at a rate of 1 mL/min. The analyses were carried out immediately after the dissolution of the polymer sample in THF to minimize sample degradation. M${}_{W}$s were calculated based on polystyrene standards. Results reported as average M${}_{W}$, percentage variation of average M${}_{W}$ (ΔM${}_{W}$), and of PI (ΔPI) were calculated after two and four weeks of incubation. ΔM${}_{W}$ and ΔPI were calculated as explained in Equations (Equation 2) and (Equation 3), respectively, where Mw${}_{0}$ is the average M${}_{W}$ of a sample before incubation, Mw${}_{t}$ is the average M${}_{W}$ of the samples at the scheduled time of incubation, M${}_{n}$ is molecular number average and $PI_{0}$ is the PI of a sample before incubation. PI${}_{t}$ is the PI of samples at the scheduled time of incubation as calculated by GPC.
(2)$$\Delta M_{w}=\frac{Mw_{0}-Mw_{t}}{Mw_{0}}\times 100$$
(3)$$\Delta PI=\frac{PI_{0}-PI_{t}}{PI_{0}}\times 100\to PI=\frac{M_{w}}{M_{n}}\times 100$$

PI from each polymer at different conditions and incubation periods were obtained. PI refers to the ratio between weight average $M_{W}$ and $M_{n}$ and indicates the distribution of polymeric chain molecular weights in a given polymer. Thus, polyester degradation involves a reduction in those individual indexes.

### 2.6. Scanning Electron Microscopy

Scanning electron microscopy (SEM) images were acquired using an Inspect^TM^ SEM F50 (FEI Company, Hillsboro, OR, USA) in an energy range between 0–30 keV. Sample preparation procedure started by a drying stage using different ethanol concentrations in water. Then, the samples were frozen separately in liquid nitrogen. Subsequently, the samples were submitted to lyophilization (Telstar cryodos Freeze Dryer). Finally, the samples were coated with a carbon film before they were examined by SEM. Porosity, pore size and surface quality were qualitatively evaluated with SEM images. Higher porosity and larger pores over time were expected due to sample degradation, as well as some morphological differences between static and dynamic conditions.

Superficial pore size distribution was quantified by measuring at least 30 pores per sample type (control, 2 weeks static, 2 weeks perfusion, 4 weeks static and 4 weeks perfusion). SEM images at 10,000× magnification were used for analysis by Image J specific tools. We compared pore size distribution of the imaged surface under different conditions in 1 μm size ranges.

### 2.7. X-ray Photoelectron Spectroscopy

Surface chemistry of printed PCL-PLGA scaffolds was analyzed with angle-resolved XPS (Spectrometer Kratos AXIS Supra, Manchester, UK). By varying the photoelectron take-off angle between 10${}^{\circ }$ and 90${}^{\circ }$ the information depth changes from about 1 to 10 nm, respectively; thus, depth profile and the surface chemistry of the topmost layer could be determined. This technique may give the following information about a sample: chemical elements identification (except H and He and if they are over 0.1%), semi-quantitative value of surface elemental composition, sample oxidation state and organic groups present. Those aforementioned parameters were summarized and represented as bound oxidation state (C-O, C-C, O-C=O) and C/O ratio. Control samples were obtained by XPS measuring non-incubated samples.

### 2.8. Mechanical Testing

To evaluate the changes in the mechanical properties over time due to degradation, a compression test was performed. Micro Tester Instron 5548 was used to obtain constant strain rates. Strain (ε)-Stress (σ) curves were obtained and non-lineal behavior was observed for high stress conditions. Thus, elastic modulus was calculated for ε=0.2. Values obtained under different conditions were compared.

### 2.9. Statistics

Each experiment was conducted 3 times. Since four samples were assayed per experiment, each condition held 12 repetitions. Incubated samples were divided at the end of each experiment to perform different analysis: 4 samples to GPC, 4 to SEM and 4 to XPS and, subsequently, mechanical testing (*n* = 4). Matlab^®^ programming language was used to run all statistical analyses. We used ANOVA test to assess significant differences between time points, and the pair-wise multiple comparison procedure was performed using the Tukey’s HSD test. A *p*-value < 0.05 was considered as significant.

## 3. Results

### 3.1. Weight Loss

Figure 3A shows the increasing weight loss over time and its enhancement under flow perfusion. Samples were weighted before sterilization at the beginning of the experiment. After the corresponding incubation period, they were dried and weighted again. Control samples underwent the same process, but they were kept at 4 ${}^{\circ }$C in a dry environment for the whole period. Table 1 summarizes the *p*-values obtained from the Tukey’s HSD test. The highest weight loss difference was found in the experiment conducted for 4 weeks under dynamic conditions, confirming the accelerated effect observed when combining longer incubation periods and flow perfusion. Static conditions involved a significant degradation after 4 weeks of incubation, and control samples also showed certain weight loss. After 2 weeks, we could not statically observe a significant weight loss in the samples regardless of the condition.

We also observed unexpected weight loss in control samples. They were kept in 24 well plates at 4 ${}^{\circ }$C with no degradation media, but neither the plate nor the fridge can ensure the optimum conditions to preserve these polymers. The variation of those may be due to the high sensitivity of PLGA to non-vacuumed environments. Water moisture adsorption degrades the polymer and would be responsible for its weight loss.

### 3.2. Incubation Medium Acidification

Exchanged PBS every 48 hours was collected for each sample and its pH measured. Environment acidification is related to synthetic polymer degradation [25]. Thus, a decrease in the PBS pH over time was expected. Figure 3B shows the pH variation up to 4 weeks of experimentation. For a better visualization of the results, no statistical lines were drawn on the plot. *p*-values comparing data at 2 and 4 weeks can be seen in Table 1.

As can be observed, significant acidification of the PBS was measured after 4 weeks of incubation, which was also enhanced by the presence of flow conditions. Continuous acidification over time was also obtained under static conditions, but the pH was maintained over 6.33 (minimum value). It can be seen that, since the beginning of the experiment, there was a medium acidification. This could be detrimental to maintain cellular viability and, therefore, it is important to take into account the potential acidification during operation. Control values were obtained by measuring fresh PBS (at 4 ${}^{\circ }$C) and, as it could be expected, no pH variation was observed.

### 3.3. Macro- and Micro-Porosity under SEM Inspection

Figure 4 shows the scaffold’s micro-porosity up to 4 weeks of experimentation. It can be qualitatively observed a superior porosity (for both incubation periods) when the scaffolds were subjected to dynamic conditions. Besides this, pores were larger at 4 weeks than at 2 weeks (Figure 5), as we hypothesized. Figure 4 also shows the macro-porosity of the scaffold that was initially printed with an established initial 3D printing space value of 400 μm. It can be noted that 3D printing achieved the requested porosity between printed fibers. This value could be easily modified by changing the different printing parameters such as the layer thickness, fiber orientation or tortuosity. This architectural tenability is one of the reasons why 3D printing is a widely used technique in BTE research.

Considering micro-porosity as a result of scaffold degradation, we could barely observe defects in the control sample fibers (Figure 4A). However, when incubated samples were imaged, micro pores on the fibers surface were observed. For the static conditions, we could observe unevenly distributed and non-continuous pores of 2–3 μm after 2 weeks. After 4 weeks, pores were slightly larger (3–8 μm) and we could observe a higher porosity. In the perfusion condition, it can be seen that the size of the pores was very similar (2–3 μm) but there was an obvious increase in the porosity compared to the ones obtained under static conditions. Finally, samples incubated for 4 weeks under dynamic conditions presented a high micro-porosity showing pores of 4–10 μm in size. Data provided were retrieved from SEM images of four samples per condition (*n* = 4).

Figure 5 quantitatively shows the pore size distribution in the samples. It can be observed that in just-printed samples imaged, the vast majority of the surface pores are <1 μm. In samples incubated during 2 weeks, we observed a superior percentage of pore appearance ranging 1–2 μm for static conditions and 2–3 μm for dynamic conditions. A similar pore size distribution was observed for samples incubated during 4 weeks under static conditions. However, the pores of 4-weeks-incubated samples under dynamic flow, presented heterogeneity. Pores measured in those samples were larger in general, even exceeding 10 μm in diameter.

### 3.4. Decrease in the Polydispersity Index

PI refers to the weight average ratio ($M_{W}$ to $M_{n}$) and indicates the distribution of the polymer chain molecular weights in a given polymer. Thus, it was expected that individual PIs of PCL and PLGA decease over time. However, due to the PCL low degradation rate, slight or no degradation was observed after 4 weeks under static conditions.

Figure 6 plots the PIs of both polymers compounding the blend at the different experimental stages. As is widely known, PLGA presents a faster degradation rate than PCL, which could be seen in our experiments. Samples incubated at 2 and 4 weeks (for both static and dynamic condition) were compared to control samples which were stored for the same time in a dry environment at 4 ${}^{\circ }$C. Some of the outcomes that were obtained from this analysis were that neither PCL nor PLGA showed significant degradation during the first 2 weeks in any of the conditions tested. In addition, 4 weeks of incubation is not enough to significantly observe PCL degradation. Nevertheless, flow perfusion implies a significant difference (*p* = 0.0495) comparing with static incubation for the PI of PCL. In the PLGA hydrolysis, both time and incubation conditions were relevant. There was a significant difference (*p* = 0.0048) between samples incubated during 2 and 4 weeks under dynamic conditions. Besides this, at 4 weeks we observed a significant decrease in the PLGA PI under flow perfusion (*p* = 0.0246). However, dynamic conditions were not significant at 2 weeks (*p* = 0.5404). Columns 5 and 6 of Table 1 summarize the statistics from these analyses.

### 3.5. Surface Elemental Analysis—Ester Bonds Hydrolysis

As already mentioned, hydrolysis breaks the ester bonds of the polymers as a degradative mechanism. This involves an increase in the number of acid groups (HO-C=O bonds). It was expected that the percentage of oxygen present would increase as a result of the polyester oxidation with a subsequent decrease in the C/O elemental ratio. Figure 7 displays elemental C/O ratios and the variation in the content of O-C=O bonds, coming from the terminal acid groups from the hydrolysis of ester bonds.

In regard to surface oxidation, a reduction in the C/O ratio was expected due to an increase in oxygen content (Figure 7B). It can be observed that samples incubated under dynamic conditions underwent a significant decrease in the ratio for both periods of incubation (*p* = 0.0023 in 2 weeks and *p* = 0.0192 in 4 weeks). These phenomena were not significant for static conditions (*p* = 0.0764 and *p* = 0.2646, respectively).

Concerning the percentage of O-C=O bond evolution (Figure 7C), statistical differences were observed between the different groups and conditions. The percentage of that bond significantly increased after 4 weeks of incubation for both static (*p* = 0.0261) and dynamic (*p* = 0.0000) conditions when compared to those of just-printed scaffolds. Besides this, there is statistical difference after 2 weeks of incubation for dynamic conditions (*p* = 0.0122), but none for static conditions (*p* = 0.9043). Comparing the groups individually, we could not observe differences in samples incubated for 2 weeks between static and dynamic assays (*p* = 0.0503). Nevertheless, after 4 weeks of incubation, there was a statistical difference between both conditions (*p* = 0.0125).

### 3.6. Mechanical Properties

Figure 8 shows the decrease in the compression elastic modulus (E) [MPa] over time. This value is the slope of the stress/strain (ε–σ) curve, which is larger when the stiffness of the material increases. Thus, a reduction in E was expected for the hydrolyzed scaffolds.

Samples incubated for 2 and 4 weeks for both static and dynamic conditions were compared to just-printed scaffolds. Results presented in Figure 8 indicate a reduction in E after 2 weeks (*p* = 0.0246) and 4 weeks (*p* = 0.0048) for dynamic conditions but no significance was observed for samples obtained under static conditions (*p* = 0.5643 and *p* = 0.1326, respectively). In Table 1, the *p*-values are compiled. The observed mechanical properties reduction was in agreement with the initial hypothesis and showed that 2 weeks are enough to weaken the scaffolds used in this study. Besides this, after 4 weeks of incubation, the mechanical properties were half-reduced.

### 3.7. Statistics

Table 1 summarizes *p*-values of each statistical analysis run in this work: % of weight loss (WL), pH variation, polydispersity index (PI) from PCL and PLGA, elastic modulus from compression tests, C/O ratio and % of O-C=O bonds. *p* < 0.05 in pink, *p* < 0.01 in green.

## 4. Discussion

Bone fractures are one of the most common organ injuries. Under healthy circumstances, bone has a unique healing capacity without scar tissue formation. However, complex bone fractures (e.g., fractures above critical size, severely damaged surrounding environment) usually fail to heal, leading to a non-union fracture. Currently, the treatment for slow or incomplete healing is bone grafting, either autograft or allograft [26]. Complications from autograft include morbidity at the harvest site, local hematoma and remodeling issues of the implanted bone. Allograft is hampered by bone tissue integration from the host and deficient vascularization issues [27]. Additionally, during bone regeneration and remodeling, there are many other factors that may cause a slower healing (either self-healing or after a graft implantation) such as infection, bone fragments micro-movements or different risk factors (e.g., smoking). Those processes may delay bone regeneration up to 9 months to achieve a total healed tissue [28].

It is also important to consider that not all bones heal equally, but it depends on their size and mechanical demand, among other factors [29]. Consequently, sustainable and long term treatment strategies are required to provide with different scaffolding to promote bone regeneration. To that end, BGS are being engineered and innovated to promote impaired fracture healing. As we mentioned before, several biomaterials have been reported as suitable materials for BGS fabrication: metals, ceramics, polymers and composites [30,31]. All of them present different advantages and disadvantages, which need to be taken into account when applied to different scenarios. For BTE, depending on the application and the defect itself, one of the most critical demands is bio-absorbability. In such manner, as new bone tissue is created, the graft loses structure, weight and mechanical properties. Mechanical solicitation is first assumed by the scaffold and, while bone healing is occurring, it will be assumed by new bone [32,33].

With all the aforementioned, it is clear that there is a need in manufacturing scaffolds to enhance bone regeneration and remodeling. Besides this, the scaffold should be engineered to fit in a specific application, depending on the bone, the zone and the defect. Thus, control over the polymeric-based scaffold degradation rates is essential, remaining as one of the inherent advantages over natural materials. Tailoring graft degradation rates and their connection to bone forming rates are essential to ensure constant mechanical properties of the damaged tissue [34,35]. All of the above make polymeric scaffolds a major category of BTE biomaterials and it is clear that thorough degradation studies are key to accurately design in vivo implants [11].

As we mentioned before, a potential technique for polymeric scaffold fabrication is electrospinning. Nevertheless, grafts final mechanical properties are low. This implies that those grafts might not be suitable for load-bearing applications [36]. Thus, the size of the defect and mechanical requirements of bone grafts are key parameters to select the manufacturing process. Three-dimensional (3D) printing emerged as promising tool to fabricate scaffolds with high precision, creating detailed 3D structures [37]. Using this technology, large bone defects geometries might be used as design patterns for scaffold fabrication [38]. There are other fabrication techniques applied to polymeric bone scaffolds: foaming methods, space holders, polymer sponges, freeze-drying and solvent casting [39]. All of those present several disadvantages regarding control over porosity and mechanical properties. Together with the aforementioned statement, we can conclude that polymeric scaffolds fabrication by 3D printing technology is a powerful tool in BTE innovation.

PCL-PLGA has been widely reported as a potential biomaterial for BGS fabrication. Peng et al. (2018) thoroughly studied cell behavior in contact with PCL-PLGA-based scaffolds of different concentrations. They determined that 50:50 wt. enhances osteogenesis in vitro [40]. It is well known that the higher the lactic content in the copolymer, the longer the degradation time span due to its hydrophobic nature. Owing to the advantages that those polyesters present, these biomaterials are a promising alternative for BTE development. Many authors have investigated potential applications of this blend in tissue engineering. Some authors have achieved a mechanical reinforcement filling the copolymer with ceramics (such as hydroxyapatite [41,42] or β-tricalcium phosphate (β-TCP)) [43]. Others have conducted some research on their application as drug delivery systems [44]. Not only for bone tissue, adaptability in mechanical properties design makes this copolymer applicable also to other tissues such as tendon [45,46], cartilage [47,48] and cardiac tissue [49].

In order to conduct a rigorous degradation study of this kind of polymers, there is a need of mimicking the physiological environment. Bioreactors allow monitoring environmental factors to provide a better understanding of the biological, chemical and physical factors involved in the healing, formation or regeneration of physiological tissue [50]. One of the strengths of these tools is providing the samples with fluid flow, which implies a more realistic recreation of the in vivo microenvironment [51], not only due to the presence of interstitial flow, but also to the mechanical stimuli produced by this flow. Shear stress together with hydrolysis are the factors that determine scaffold degradation [52].

In this work, hydrolytic degradation of 3D printed PCL-PLGA scaffolds was thoroughly characterized. Despite lacking certain in vivo conditions, physiological parameters were considered thanks to the use of a customized biorreactor. This tool allowed the application of a perfusion flow to the samples, emulating interstitial flow in bone tissue in vivo. Previous bioreactors have been customized for specific experiments. Bhaskar et al. (2018) designed a system hosting one sample for the study of large bone defects (up to 30 mm in diameter) [53]. They also corroborated the importance of the fluid flow in terms of cell proliferation throughout the scaffold and osteogenic differentiation of human embryonic stem cell-derived mesenchymal progenitors. The bioreactor presented here offers superior versatility in terms of interfacing, samples connectivity and flow rates. Depending on the experiment purposes and requirements, a different connecting tubing set up could have been designed. Other bioreactors have been developed, but they lack the versatility of the system and also the easy handling of the one here reported. Dimensions of 10 cm long, 2 cm wide and a total weight of 90 g make the whole system totally portable and allows the use of small sample sizes, which is important when investigating costly materials. This fact together with the possibility of sterilizing all components by autoclaving, allow the user to easily introduce the system inside an incubator. All the above makes our tool suitable for cell culture experiments.

Our results showed a significant weight loss of the polymeric scaffolds over time, especially when samples were incubated under dynamic conditions. This directly correlates with a loss in the mechanical properties of the scaffold: it was confirmed that the reduction of the compression elastic modulus is associated with a structural weakening due to its faster degradation. We observed that degraded samples under dynamic conditions after 4 weeks of incubation presented the lowest values of compression elastic modulus. The reason for the weight and mechanical stiffness loss over time is caused by the known hydrolysis of the polyester. SEM images also showed qualitatively and quantitatively the evolution of the micro-porosity of the scaffolds and confirmed the aforementioned outcome. Predictably, after a longer implantation period in vivo, mechanical properties loss would be probably greater. Nevertheless, as mentioned before, PLGA degrades faster than PCL, which has been shown to persist in the body after 2 years of implantation [54], and therefore, once PLGA is completely degraded, the scaffold’s degradation rate would be that of PCL. Thus, it could be predicted that at this point the scaffold’s weight loss would be slowed down.

Bone elastic modulus is a variable value depending on the bone itself and its nature (trabecular or cortical). For instance, the femur has been widely studied and its elastic modulus (for traction and compression) is between 10 and 20 MPa [55]. Previously, other authors customized scaffolds made of different materials and had their mechanical properties measured. Barui et al. (2017) created scaffolds by 3D printing Ti-6Al-4V powders. They obtained an elastic modulus between 2–7 GPa and compressive strength between 90–150 MPa [56]. Xu et al. (2017) reported a procedure to fabricate chitosan scaffolds with tunable mechanical properties. The stiffest sample had a compression elastic modulus of 25 KPa [57]. Just-fabricated scaffolds reported here presented an elastic modulus mean value around 5 MPa, which is close to the magnitud of the bone. Thus, it is also confirmed the benefit of using polymers in BTE. Hence, 3D printing technical feasibility together with the use of biodegradable polymers allow the fabrication of customized scaffolds in terms of geometry and degradation rate. This fact is crucial to engineer a graft consistent with the mechanical demand of a specific bone area.

Beholding degradation chemistry, ester bond scission is the main mechanism of the degradation of polyesters in vitro. At first, the macro-molecular chains are hydrolyzed into water-soluble oligomers and monomers, and then they are released into the surrounding medium [58]. Besides this, the degraded acidic products may accelerate the continuous hydrolysis process. As we mentioned before, PLGA is a poly(α-hydroxyl-ester) that can be depolymerized in the presence of water and is more prone to be hydrolyzed than PCL [59]. In this work, % of O-C=O bonds and C/O ratio in the scaffold surface were obtained as indicators of surface oxidation (Figure 7), directly related to their hydrolytic degradation [60]. This parameter for each sample was compared along different conditions and incubation periods. We also monitored medium acidification over time by measuring the pH of replaced PBS (Figure 3B). The literature shows that medium acidification is caused by the hydrolytic process, where ester bonds are hydrolyzed under acidic conditions forming the corresponding parent carboxylic monomers, namely, lactic and glycolic acids [61]. Thus, an increase in the oxygen content was expected. As it can be observed in Figure 7B,C, those outcomes are in agreement: the medium pH reduction over time matches with a decrease in superficial C/O ratio and with an increase in the amount of O-C=O bonds. The severe medium acidification that we observed could risk cellular viability over time in cell culture experiments. However, in the presented study, cells were not used and PBS was exchanged every 48 h. The possibility of monitoring the medium pH every day makes it possible to ensure a viable environment in case of holding a cell culture. For instance, it could be possible to exchange the culture medium frequently. Moreover, regarding a potential clinical use, in bone implantation, there would be a continuous flow exchange which would not dramatically compromise the pH of the area.

The thorough degradation study presented here involves a full and detailed analysis that might help to understand the mechanisms behind polyester-based grafts degradation under static and dynamic conditions. Being aware of this information, it could be possible to design bone scaffolds with a customized degradation rate by combining specific amounts of PLGA and PCL. PLGA degradation depends on the ratio of its composing monomers, on the molecular weight of the polymer and on the ester or free carboxyl end groups. The higher its molecular weight, the longer its degradation timing. In general, the higher the glycolic acid content in the copolymer, the faster its degradation rate because the superior hydrophilic character of glycolic repeat units compared to lactic acid ones results in a greater degree of water uptake during hydrolysis. However, PLGA 50:50 shows the fastest degradation rate compared to other PLGA copolymers, even those having higher glycolic acid content.

Although several studies in the literature have investigated the in vitro hydrolytic degradation of polyesters, the study here reported offers quantitative data under perfusion conditions, which closely mimics the physiological interstitial perfusion that occurs in vivo. Ultimately, perfusion flow creates shear forces that are mechanotransduced by the cells to exhibit specific cellular responses to promote bone formation. Therefore, we think that it is critical to include flow perfusion in any experimental set up that evaluates degradation and integration. Ideally, under operation, the scaffold would degrade at the same speed that cells colonize it. Our proposed scaffolds having two different blended polymers would also allow fine-tuning the degradation rate of newly developed polyester-based scaffolds.

Finally, our bioreactor can hold assays with no time limits because of the possibility to easily exchange culture medium. Besides this, owing to its size and easy handling, it is possible to conduct several experiments in parallel. Thus, this bioreactor could accommodate different experiments: various kinds of biomaterials (either other blends or even metals or ceramics), longer incubation periods, which would mimic a non-union, and different flow perfusion rates, among others.

## 5. Conclusions

In this work a thorough degradation characterization of PCL-PLGA (50:50) 3D printed scaffolds was conducted. We performed a quantitative degradation study in terms of morphology, chemistry and mechanical properties. Obtained results are in agreement with our hypothesis: PLGA degrades faster than PCL and flow perfusion is critical in the degradation process producing an accelerated hydrolysis and up to 4 weeks are needed to observe significant degradation in polyester scaffolds of this size.

Our customized in-house fabricated bioreactor allowed the conduction of degradation experiments. The shear stress induced by the flow perfusion and the possibility of maintaining the bioreactor in a controlled-environment incubator allowed to mimic several of the features of the physiological environment. All of these facts, make the bioreactor highly valuable to monitor degradation and it may become a powerful tool in scaffolding research for BTE.

Future work in this topic intend to introduce a cellular element to the scaffold to better recreate the in vivo environment that the scaffold would encounter upon implantation. The bioreactor and polyester features grant culture biocompatibility. That, together with a continuous flow of culture medium, would allow culture viability.

## Figures and Tables

**Figure 1 materials-15-02572-f001:**
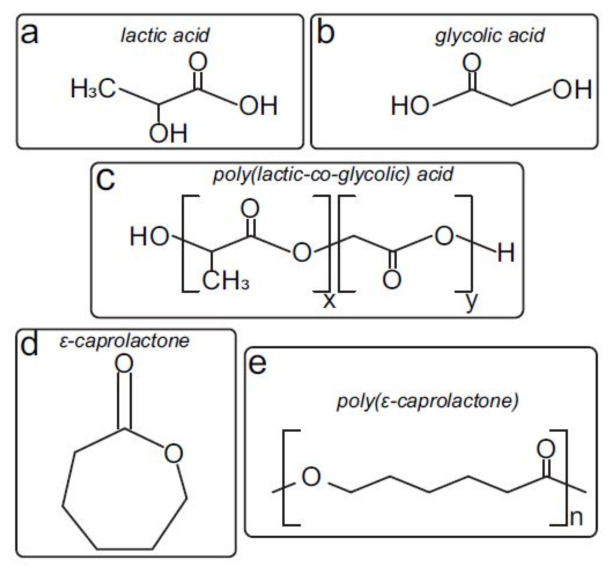
Structural chemical formulas of (**a**) lactic acid, (**b**) glycolic acid, (**c**) poly(lactic-co-glycolic) acid, (**d**) ε-caprolactone and (**e**) poly(ε-caprolactone). PLGA and PCL degrades via chain scissions of ester bond linkages in the polymer backbone (**c**,**e**) by hydrolytic attack of water molecules.

**Figure 2 materials-15-02572-f002:**
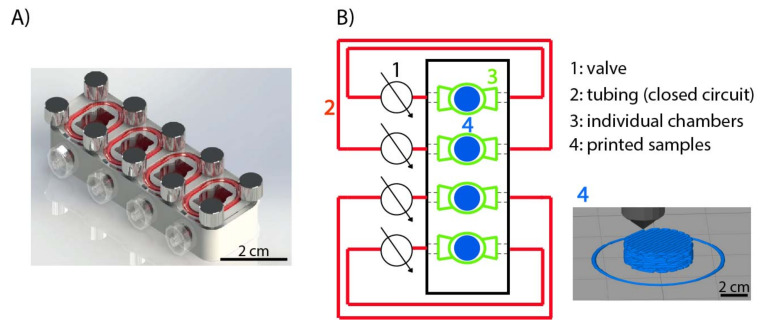
In-house fabricated bioreactor used for study of degradation under static and dynamic conditions. (**A**) Render of the bioreactor used in this study. (**B**) Simplified scheme of the bioreactor where the elements that compose the bioreactor system are: (1) valves, (2) tubing system which allows four different closed circuits, (3) bioreactor individual chambers, (4) sample along with a schematic of a 3D printed sample.

**Figure 3 materials-15-02572-f003:**
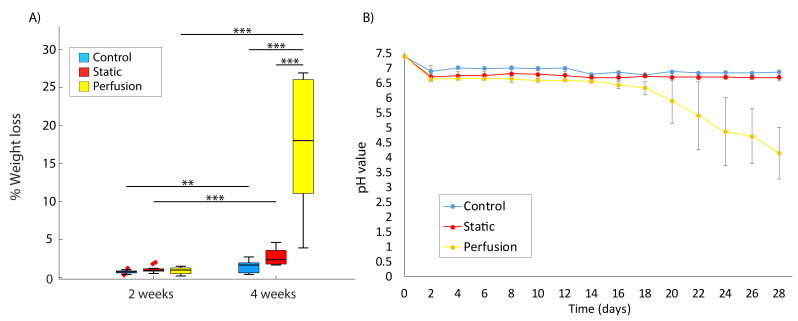
(**A**) Weight Loss (%) of samples after 2 and 4 weeks of incubation for the different conditions: control (blue), static (red) and perfusion (yellow). (**B**) pH variation over 4 weeks of incubation. Control values (blue) are taken by measuring PBS at 37 ${}^{\circ }$C every two days. Values from static (red) and perfusion (yellow) conditions were taken by measuring the collected exchanged PBS every 48 h. Significant different were assumed for a *p*-value < 0.05 (** *p* < 0.005, *** *p* < 0.001), x outlier point.

**Figure 4 materials-15-02572-f004:**
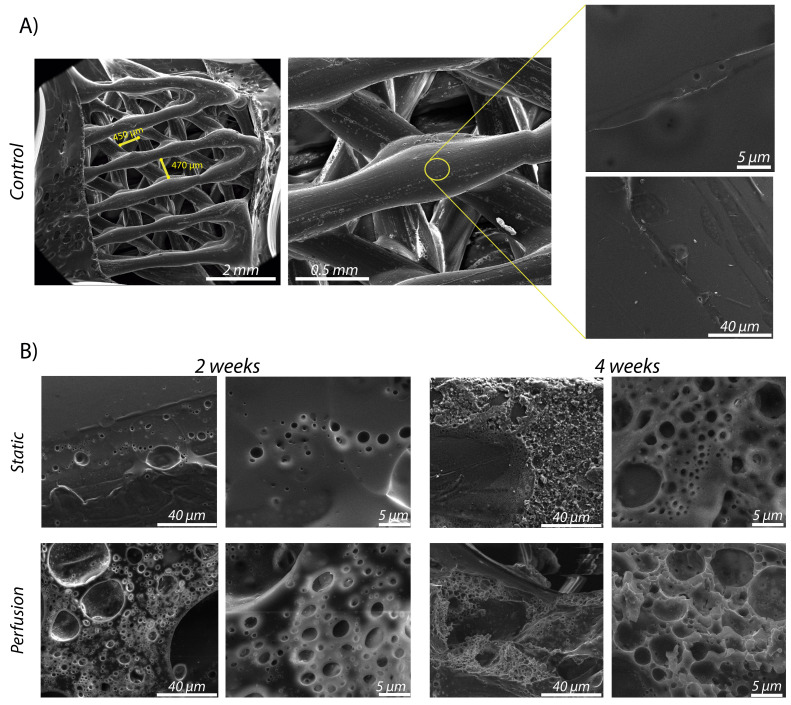
Representative SEM images: (**A**) view of macro porosity and 2500× and 10,000× magnification of a non-incubated scaffold (**B**) samples incubated for 2 and 4 weeks under different conditions: static and dynamic (at 2500× and 10,000× magnification).

**Figure 5 materials-15-02572-f005:**
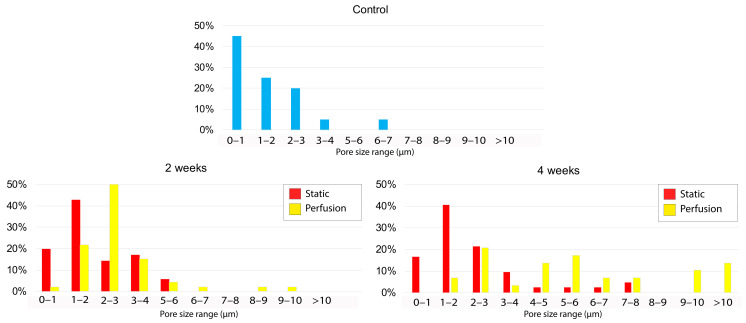
Surface pore size distribution. Percentage of pore size occurrence in the different samples: control (blue), static (red) and perfusion (yellow). At least 30 pores were measured for each condition to determine size distribution.

**Figure 6 materials-15-02572-f006:**
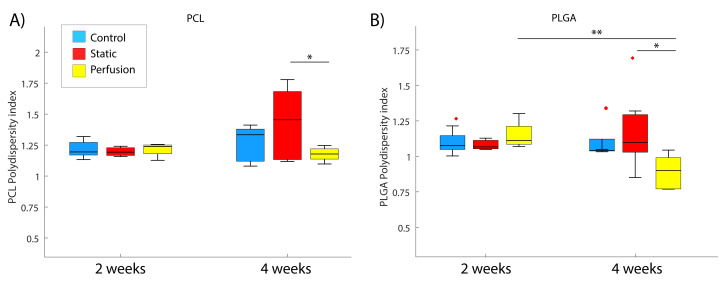
Gel permeation chromatography results: (**A**) PCL and (**B**) PLGA -polydispersity index of samples after 2 and 4 weeks of incubation for the different conditions: control (blue), static (red) and perfusion (yellow). PIs were quantified as Mw/Mn ratios of different samples. Measurements from control samples were done at the same time points but samples did not undergo any incubation. Statistical difference considered for a *p*-value < 0.05 (* *p* < 0.05, ** *p* < 0.005), x outlier point.

**Figure 7 materials-15-02572-f007:**
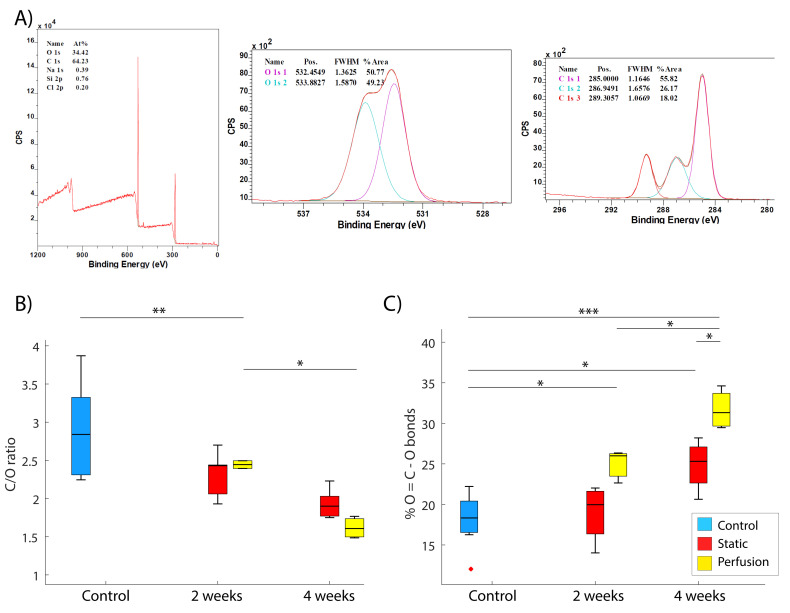
(**A**) XPS report: general spectrum of an analyzed zone (300 μm) where the main signals are obtained for carbon and oxygen (left) and high resolution spectrum for specific regions of C and O (1 s) (right). (**B**) C/O ratio on the sample surface. (**C**) % of O-C=O bonds in the samples. Readings were obtained in the first 10 nm of the surface. For both B and C plots: Control data (blue) was obtained from non-incubated scaffolds. Static (red) and perfusion (yellow) were assayed after drying the sample at the end of the experiment. Statistical significance was considered for a *p*-value < 0.05 (* *p* < 0.5, ** *p* < 0.005, *** *p* < 0.001), x outlier point.

**Figure 8 materials-15-02572-f008:**
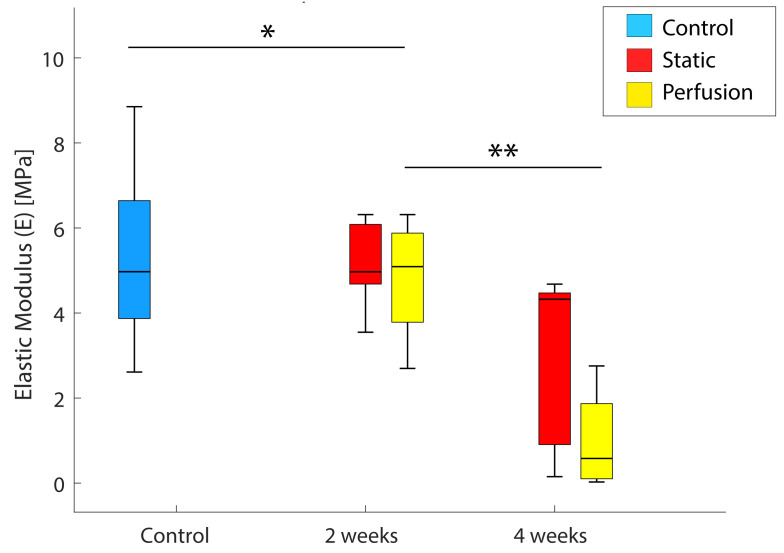
Compression elastic modulus variation (MPa) of samples after 2 and 4 weeks of incubation under static (red) and perfusion (yellow) conditions. Samples without incubation period were selected as control (blue). Statistical significance was considered as *p*-value < 0.05: * *p*-value = 0.0305 (*p* < 0.05), ** *p*-value = 0.0048 (*p* < 0.005).

**Table 1 materials-15-02572-t001:** *p*-values of statistical analysis run in this work: % of weight loss (WL), pH variation, polydispersity index (PI) from PCL and PLGA, elastic modulus from compression test, C/O ratio and % of O-C=O bonds. *p* < 0.05 in pink, *p* < 0.01 in green. ANOVA test was used to assess significant differences between timepoints and the pair-wise multiple comparison procedure was performed using the Turkey’s HSD test.

Conditions	Weeks	p WL	p pH	p PI${}_{\mathrm{PCL}}$	p PI${}_{\mathrm{PLGA}}$	p E	p C/O	p -OH
Control–control	2-4	0.0034	0.7657	0.6900	0.9579	-	-	-
Static–static	2-4	0.0000	0.3654	0.1903	0.5643	0.1326	0.9390	0.0789
Perfusion–Perfusion	2-4	0.0000	0.0000	0.2593	0.0048	0.0048	0.0192	0.0232
Control–static	2-2	0.4155	0.0000	0.8278	0.9112	0.3863	0.0764	0.9043
Control–perfusion	2-2	0.0543	0.0000	0.8287	0.6164	0.0305	0.0023	0.0122
Static–perfusion	2-2	0.5389	0.0296	0.9984	0.5404	0.3863	0.2736	0.0503
Control–static	4-4	0.7688	0.8205	0.3174	0.7820	1.000	0.2646	0.0261
Control–perfusion	4-4	0.0000	0.0000	0.6988	0.1621	0.1182	0.0028	0.0000
Static–perfusion	4-4	0.0000	0.0000	0.0495	0.0246	0.1494	0.1587	0.0125

## Data Availability

The data that support the findings of this study are available from the corresponding author upon reasonable request.

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
