# Peer review of "In Vitro Hydrolytic Degradation of Polyester-Based Scaffolds under Static and Dynamic Conditions in a Customized Perfusion Bioreactor"

_materials, 2022, doi:10.3390/ma15072572_

Round 1

Reviewer 1 Report

The manuscript reported the PCL-PLGA composites for as scaffolds. The concept is not new since similar system have been reported before. Hydrolytic degradation of polyester is not the only issues used for scaffolds. There are so many developed and reported materials still have many big problems after evaluated by various hydrolytic degradation under static and dynamic conditions. All the applications have to be tested and evaluated by clinical experimental. Authors mentioned that flow perfusion is critical in the degradation process of PCL-PLGA based scaffolds implying an accelerated hydrolysis compared to the ones produced under static conditions. It is a comment sense and predicable. The key issue is  how to avoid the flow perfusion? What is the answer to solve this challenge?

Reviewer 2 Report

The authors have presented a detailed study that examines the effects of perfusion flow bioreactor on the degradation process of 3D-printed PCL-PLGA scaffolds. All experiments were explained in detail and the results corroborate their hypothesis. The portability and customizability of the bioreactor setup is greatly appreciated.

  1. Samples used as controls should be mentioned explicitly in the beginning, either in the introduction or in the methods sections. It required some reading back and forth to understand that control samples were the same formulation, that were not exposed to PBS.
  2. If control samples were not subjected to any degradation media, how do they have a mass loss after 4 weeks as seen in Figure 3A?
  3. Figure 3B shows a large reduction in pH of the surrounding media after 4 weeks of degradation under a perfusion flow setup. This drop in pH could be toxic to the surrounding bone tissue during implantation. How does the author justify use of PCL-PLGA scaffolds that exhibit this pH reduction?
  4. The authors mention that perfusion setup with continuous flow exhibits a significant mass loss after 4 weeks. However, the total mass loss amounts to 0.26% according to Figure 3A. Kindly explain the significance of this mass loss on the overall characterization of the dynamic flow system. Also, it is difficult to yield such precision to measure 0.26% average mass loss across 12 samples using a weighing scale.
  5. If the elastic compression modulus is halved within 4 weeks while undergoing a 0.26% mass loss, how would the mechanical properties be affected after a higher mass loss over a longer implantation period?

Reviewer 3 Report

This is a well-written, comprehensive study on a novel system for scaffold biofabrication and characterization. It would be ideal if the authors could add some cell culture dat, as they discussed in the article but it is not a must as it would change the manuscript focus. I will be looking forward to their next study with cells on this. I only have 2 suggestions for this manuscript:

Figure 5-missing error bars.

Adding n number for analyses might be useful.

Thank you

Round 2

Reviewer 1 Report

The revised manuscript should be considered as accept.  

Reviewer 2 Report

The authors have adequately addressed all review comments.